# Designing for Equity: An Evaluation Framework to Assess Zero-Dose Reduction Efforts in Southern Madagascar

**DOI:** 10.3390/vaccines13080834

**Published:** 2025-08-05

**Authors:** Guillaume Demare, Elgiraud Ramarosaiky, Zavaniarivo Rampanjato, Nadine Muller, Beate Kampmann, Hanna-Tina Fischer

**Affiliations:** 1Center for Global Health, Charité—Universitätsmedizin Berlin, 10117 Berlin, Germany; 2Doctors for Madagascar, Logt II M 98 G Antsakaviro, Antananarivo Appt III-12, Madagascar; 3Department of Infectious Diseases and Critical Care Medicine, Charité—Universitätsmedizin Berlin, 10117 Berlin, Germany; 4Berlin Institute of Health, Charité—Universitätsmedizin Berlin, 10117 Berlin, Germany

**Keywords:** zero-dose children, immunization, equity, evaluation

## Abstract

Despite growing global momentum to reduce the number of children who never received a dose of any vaccine, i.e., zero-dose (ZD) children, persistent geographic and social inequities continue to undermine progress toward universal immunization coverage. In Madagascar, where routine vaccination coverage remains below 50% in most regions, the non-governmental organization Doctors for Madagascar and public sector partners are implementing the SOAMEVA program: a targeted community-based initiative to identify and reach ZD children in sixteen underserved districts in the country’s south. This paper outlines the equity-sensitive evaluation design developed to assess the implementation and impact of SOAMEVA. It presents a forward-looking evaluation framework that integrates both quantitative program monitoring and qualitative community insights. By focusing at the fokontany level—the smallest administrative unit in Madagascar—the evaluation captures small-scale variation in ZD prevalence and program reach, allowing for a detailed analysis of disparities often masked in aggregated data. Importantly, the evaluation includes structured feedback loops with community health workers and caregivers, surfacing local knowledge on barriers to immunization access and program adoption. It also tracks real-time adaptations to implementation strategy across diverse contexts, offering insight into how routine immunization programs can be made more responsive, sustainable, and equitable. We propose eight design principles for conducting equity-sensitive evaluation of immunization programs in similar fragile settings.

## 1. Introduction: Zero-Dose Children

Despite decades of global immunization efforts, millions of children still remain unreached by routine vaccination services [1,2]. Among these are penta-zero dose (ZD) children—zero dose being defined as “non-receipt of the first dose of the DTP-containing pentavalent vaccine (i.e., DTP1)”, which protects against diphtheria, tetanus, pertussis, hepatitis B, and *Haemophilus influenzae* type B [3]. These children face heightened vulnerability to vaccine-preventable diseases and are typically found in settings marked by multiple deprivations, including poverty, gender inequities, and geographic isolation [4,5,6]. The Immunization Agenda 2030 (IA2030) and the zero-dose strategy articulated by Gavi, the Vaccine Alliance in 2021, have both placed ZD children at the center of immunization priorities, aiming to reduce prevalence by 50% by 2030 across 54 eligible countries. Because ZD prevalence reflects deeper issues of health system exclusion [7,8], addressing those is integral to the broader Sustainable Development Goals (SDGs), especially in achieving universal health coverage (SDG 3.8). As a result, there is a growing recognition that strategies to reduce ZD prevalence must be tailored, data-driven, and equity-sensitive [1].

Madagascar, one of the countries in the world with highest levels of multidimensional poverty [9], exemplifies these challenges. Despite adopting the WHO Expanded Program on Immunization (EPI) in 1976, national routine vaccination coverage remains among the lowest globally due to systemic challenges in health service delivery. ZD prevalence exceeds 50% in most regions, especially in the South of the country [10]. Contributing factors identified are geographic isolation, low community-level trust and awareness, lack of personnel in health facilities to provide immunization services, vaccine stockouts, equipment shortages, and inequity [11,12].

In response, the non-governmental organization Doctors for Madagascar (DFM), in collaboration with Gavi, as well as regional and national health authorities and public sector healthcare providers, launched SOAMEVA (“SOrohy ny Aretina Manaova Ezaka VAksiny”, which translates to “Prevent disease, make an effort to get vaccinated”) in 2024. It is a community-based program operating at the fokontany level—the lowest administrative unit in Madagascar. The program is targeting all 16 districts of Madagascar’s three southernmost regions, Atsimo-Andrefana, Androy, and Anosy, which are characterized by extreme rates of poverty, extreme weather events, and severe food insecurity [13,14]. The overarching aims of SOAMEVA are to reduce the number of ZD children by 40,000 (from an estimated 50,000, i.e., 80%) in those regions and to strengthen the immunization delivery capacity of local health facilities (CSBs), in collaboration with community health workers (CHWs) and district health authorities.

This perspective paper outlines the equity-sensitive evaluation strategy developed for SOAMEVA, using the RE-AIM (Reach, Effectiveness, Adoption, Implementation, and Maintenance) and PRISM (Practical, Robust, Implementation, and Sustainable Model) frameworks [15,16]. We focus here on the rationale, methodology, and planned analytical approach for the evaluation of the SOAMEVA program.

## 2. The Context: The SOAMEVA Program

To inform program design, the different SOAMEVA components were co-created with stakeholders across all administrative levels via a rapid assessment conducted by DFM to identify priority needs, challenges, and baseline conditions across the three regions. This included interviews with district-level Medical Officers (médecins inspecteurs) and EPI Managers (ResPEV) of all 16 districts, a service availability and readiness assessment of vaccination services at all CSBs in the catchment area (N = 375), and interviews with CHWs, community leaders, and caregivers, conducted in fokontany with contrasting ZD prevalence. Additionally, the baseline assessment included a secondary data analysis of preliminary findings from a parallel mixed-methods study on the characteristics and motivations of CHWs [17], as well as insights from key informant interviews with local community health experts. As the different SOAMEVA components were embedded within regional and district-level immunization plans, they took into consideration local contexts and needs with regard to ZD children in southern Madagascar.

One key component of the SOAMEVA program, launched in September 2024, is a full-scale community census conducted in over 4000 fokontany to identify ZD children under five years of age and to estimate ZD prevalence at the lowest administrative level. Implemented by trained CHWs, the census cross-verified household data, community registries, and immunization records at the CSB level. Children with no documented evidence of having received any dose of DTP-containing vaccine were classified as zero-dose (i.e., penta-ZD), as per the definition used by Gavi [3]. Gathering accurate ZD data is not only crucial for effectively guiding the different SOAMEVA activities, especially for targeting fokontany with the highest numbers of ZD children, but also to inform the national EPI, given that national immunization strategies are dependent on reliable estimates of ZD prevalence. For example, the dispatching of vaccines to local health centers is calculated based on those estimates. Inaccurate ZD estimates can result in misallocation of resources and insufficient vaccine supply where it is most needed.

Building on the census activity, the SOAMEVA program closely collaborates with CSBs and CHWs to conduct joint mobile vaccination clinics in fokontany with the highest numbers of ZD children. For a subset of those fokontany, the program also includes a community sensitization campaign via mobile cinema (i.e., VacCiné) preceding the vaccination sessions. In addition, SOAMEVA supports CSBs through cold-chain repair and maintenance training, logistical support to prevent vaccine stockouts, and needs-based provision of essential EPI tools, such as vaccination cards and registries. The SOAMEVA program components, as well as their intended outcomes, are detailed in Figure 1. The SOAMEVA inputs include financial incentives for CHWs to conduct the community census and mobile outreach activities, the provision of vehicles for all program operations, staff to conduct the activities (e.g., training of CHWs), and financial support to strengthen the capacity of CSBs. Overall, the SOAMEVA program is expected to reach fokontany with the highest number of ZD children and reduce their prevalence in the intervention zone, both directly via mobile vaccination clinics, and indirectly by addressing community hesitancy and strengthening the capacity of CSBs to deliver immunization services, i.e., the national EPI.

By collecting fine-scale information on ZD data, and by aligning its different components with local needs, the SOAMEVA program intends to reduce the equity gap surrounding routine immunization. This disaggregated approach allows for precise targeting of its different activities, where aggregated district or regional information would otherwise mask disparities. While certain components of SOAMEVA, such as mobile vaccination campaigns, follow a vertical approach aimed at directly reducing the number of ZD children, other components, including strengthening the capacity of CSBs and increasing community vaccine-seeking behavior, have the potential to contribute to broader health system strengthening. This dual approach reflects an increasing emphasis in global health on integrating targeted service delivery within longer-term health systems, strengthening efforts, e.g., [18,19]. A first dose of DTP-containing vaccine can reduce the risk of contracting life-threatening diseases such as pertussis by ca. 50% [20]. However, long-lasting immunity for children requires completing the full vaccination schedule (i.e., three doses), which SOAMEVA aims to achieve by visiting each target fokontany three times. Ultimately, however, the most substantial health outcomes may only be achieved by strengthening the underlying health system and ensuring equitable and continuous access to immunization services for the most vulnerable populations.

## 3. Embedding Equity in Evaluation Design

The SOAMEVA evaluation is rooted in the RE-AIM and PRISM frameworks [15,16]. While RE-AIM offers a structured yet flexible framework to measure both intervention processes and outcomes, PRISM is an analytical lens that allows us to explore contextual factors influencing the intervention, both from a recipient (e.g., community members) and organizational (e.g., SOAMEVA program leaders, CSBs) perspective. Here, each RE-AIM dimension explicitly integrated equity-sensitive aspects, as detailed below. Each element of the program is assessed not only in terms of what it delivers, but for whom, where, and with what implications for equity (Figure 2).

**Reach**. *To what extent did the different activities reach the target fokontany and target population?* The granular ZD and programmatic data collected by DFM—a major source of secondary data for the evaluation—will allow us to measure disparities in program reach between fokontany with respect to ZD prevalence. In addition, a follow-up survey at the end of the project period for a subset of 250 fokontany will allow us to measure net pre-post changes in ZD prevalence at the fokontany level. Qualitative data collection will also allow us to identify key factors that account for disparities in program reach.

**Effectiveness**. *To what extent did the program affect both supply and demand of routine vaccination services?* While examining effectiveness components, such as the reduction in number of ZD children through the SOAMEVA program, the evaluation will attempt to measure unintended consequences, such as strain on health worker capacity. Looking at the differences in outcomes between fokontany, we will attempt to better understand potential equity-related variation in program effectiveness.

**Adoption**. *To what extent did CHWs, CSBs, and community members engage with program activities?* While many program components rely on active participation, especially from CHWs, it is possible that differences in participation may impact both program reach and effectiveness. As part of the evaluation, we aim to identify equity-related barriers and facilitators for engagement.

**Implementation**. *How consistently, and with what adaptations, was the SOAMEVA program implemented, and what were the time and cost implications?* A key characteristic of SOAMEVA is its willingness to align with local context and priorities, which can change throughout the intervention period. As such, an important component of the evaluation will be to determine what adaptations were made, and to what extent those helped maximize equitable reduction in ZD prevalence.

**Maintenance**. *To what extent does the SOAMEVA program represent a health system strengthening approach contributing to sustainable improvements in routine immunization capacity?* While SOAMEVA offers great potential for strengthening the existing health system, including CSBs and CHWs, local conditions may either facilitate or hinder maintenance beyond the program timeline. Measuring the potential for maintenance, and associated equity-relevant factors, is an important component of the evaluation.

Overall, the evaluation relies on disaggregated data across all key indicators. Fokontany-level analyses are used not only to assess program performance but also to measure small-scale variation in both ZD burden and program reach. This level of resolution is critical in fragile settings where disparities often result from geographic, infrastructural, or social exclusion [7]. The fokontany level was selected as the unit of analysis because it is the smallest administrative unit in Madagascar with clearly defined geographic boundaries, population data, and health service linkage. This level of granularity is essential for detecting small-scale variation in ZD prevalence and program reach—variation that is often masked at higher administrative levels. In Box 1 we propose eight points to take into consideration when designing the equity-sensitive evaluation of immunization programs.

Box 1Design Principles for Equity-Sensitive Evaluation of Immunization Programs.
Use disaggregated data—Analyzing
data at a fine scale helps to reveal hidden disparities in ZD prevalence and
program reach.Embed equity into frameworks—Using
frameworks such as RE-AIM and PRISM not just to assess performance but also
to surface underlying inequities.Pair numbers with
narratives—Combining quantitative coverage data with qualitative insights at
the community level helps to identify barriers.Design for adaptation—Building in
structured feedback loops allows real-time adjustments to program delivery in
response to early evaluation findings.Center local perspectives—Treating
CHWs and caregivers as co-producers of learning, not just respondents, helps
to integrate community knowledge.Define and measure
marginalization—Terms like “hard-to-reach” can be operationalized using
concrete indicators (e.g., distance to health center).Plan for equity before measuring
impact—Integrating equity into the evaluation begins in the design phase, not
only after results are obtained.Include feedback loops—Collecting
perspectives from recipients of the intervention and sharing findings with
local stakeholders, partners, and communities to promote transparency and
accountability.


## 4. A Mixed Methods Approach

The SOAMEVA evaluation adopts a mixed-method approach, integrating quantitative data on ZD prevalence for each fokontany (including a follow-up census for a subset of 250 randomly selected fokontany at the end of the intervention period), CSB readiness, and vaccination session logs. The outcomes and processes measured quantitatively will be contextualized through a qualitative investigation of different actors involved in the SOAMEVA program via semi-structured key informant interviews, as well as focus group discussions with community members. Participants for qualitative data collection, including CHWs, caregivers, and community members, will be purposively selected to reflect diverse geographic and programmatic contexts, with data collection continuing until thematic saturation is reached. To mitigate social desirability bias, data collectors will be external to both the SOAMEVA program and Madagascar health system and will be trained in neutral interviewing techniques. Interviews will be conducted in private settings.

The quantitative analysis will not only allow us to measure key indicators of outcomes and processes underlying the SOAMEVA program, such as absolute reductions in number of ZD children through mobile vaccinations, but it will also enable the identification of a subsample of fokontany to conduct targeted focus group discussions. For example, we plan to qualitatively compare fokontany that achieved the greatest reductions in ZD prevalence (i.e., positive deviants) with those where SOAMEVA may not have had as high an impact, to identify key factors of intervention success. Positive and negative deviants (*n* = 8, respectively) will be identified by ranking fokontany based on z-scores of relative reductions in ZD prevalence, measured post-SOAMEVA activities (i.e., mobile clinics).

## 5. Designing for Real-Time Adaptations

One important aspect of the SOAMEVA evaluation is to provide midpoint insights to DFM and partners, including regional- and district-level authorities, to reflect on all RE-AIM dimensions as the program is being implemented, which may lead to a refinement of intervention delivery. As such, the evaluation integrates structured feedback loops with CHWs, who play a key role in the delivery of multiple SOAMEVA components, such as mobile vaccinations, as well as community members, to collect feedback as recipients of the intervention. This part of the evaluation will provide key insights into program delivery, including the identification of important barriers to implementation, and potential unintended consequences of the intervention. For example, vertical CHW-driven HIV programs in South Africa that were not adequately embedded in local community settings led to unintended consequences, including CHWs moonlighting for multiple organizations and inadequate care for patients due to uncoordinated care [21].

While full implementation of the evaluation is forthcoming and was not designed in conjunction with the planning of the intervention, it was intentionally designed to enable course correction in response to emerging findings. This includes structured feedback loops with CHWs, adaptive microplanning at the fokontany level, and collaborative reflection sessions with CSBs and district authorities. These mechanisms aim to detect early equity gaps, such as variation in caregiver attendance or CHW engagement, and adjust outreach or supply chain strategies accordingly. The design anticipates that “learning while doing” is critical to addressing inequities that may not be visible in pre-intervention planning [16,22], especially given the complexity and scale of SOAMEVA. This adaptive approach is particularly valuable in fragile settings, where intervention plans must consider local variability, logistical barriers, as well as unforeseen challenges. For example, the regular monitoring of CSB capacity and vaccination activities, combined with qualitative data collections at the CSB and district levels, will enable the identification of important bottlenecks (e.g., transport) and collect organizational perspectives at different levels to identify solutions that can be communicated with DFM. In addition, exploring community perspectives may reveal important equity-sensitive disparities in program reach and effectiveness, such as the ability of CHWs and community members to engage with SOAMEVA as a result of sociodemographic factors. Finally, an important part of the evaluation will be to qualitatively assess the organizational capacity in integrating data-driven adaptations. If successfully integrated, program adaptations can play a significant role in addressing equity gaps.

## 6. Evaluating for Equity in Fragile Settings

Designing an equity-sensitive evaluation in fragile health systems requires navigating trade-offs. The SOAMEVA evaluation integrates a dual focus on targeted ZD reduction and longer-term systems strengthening, as reflected in the design of the SOAMEVA program itself. Yet we anticipate challenges. While mobile vaccination clinics may effectively reach a large number of ZD children, sustaining impact will ultimately depend on the strength of the health system in place [23]. We also expect variation in CHW engagement, shaped by geography, incentives, and competing responsibilities. By embedding mechanisms to surface and respond to such dynamics, the evaluation reflects a growing consensus that measuring equity cannot be an afterthought and must instead be built into the architecture of the evaluation itself.

The SOAMEVA program, in its design, set the ground for developing an equity-sensitive evaluation framework (see Box 1), especially by collecting ZD prevalence data throughout the intervention zone at a fine spatial resolution, allowing targeted intervention and measuring program outcomes. Using disaggregated data on ZD prevalence is often a major challenge in fragile settings such as Madagascar, where absence of or inconsistency in documentation on immunization coverage may result in significantly biased estimates, and yet is critical to address equity-sensitive aspects of programs aiming to reach ZD children. By triangulating evidence from quantitative reductions in ZD prevalence and qualitative investigations, the evaluation is expected to reveal not only where inequalities persist but also why.

In low-to-middle income countries, under-vaccination is commonly driven by structural barriers such as geographic isolation and weak health systems [24]. Other studies investigating fragile contexts, such as Ethiopia and the Democratic Republic of Congo, have found key determinants of ZD prevalence similar to those previously identified in the Madagascar context, and have therefore emphasized the need for more equitable access to immunization services [25,26,27]. In addition, there is a growing body of evidence demonstrating the importance of CHW-led interventions to improve access to health services for marginalized groups in low-to-middle income countries [28,29]. While context-specific, the design of the SOAMEVA evaluation, with its core emphasis on the role of CHWs, has the potential to be adapted to other fragile settings to enhance equitable access to immunization efforts.

Ultimately, equity-sensitive evaluation requires shifting the role of evaluation itself: from a passive record of performance to an active driver of inclusion. As such, the SOAMEVA evaluation was designed as a tool for change, rather than simply measuring program outcomes at a broad scale. The focus is on the smallest unit of intervention: the community. In doing so programs become better able to effectively reach children who are consistently missed, hence narrowing the equity gap in immunization.

## 7. Conclusions

In fragile and under-reached settings, embedding equity in evaluation is not only a moral imperative but also a practical necessity. This perspective paper introduces a replicable, equity-driven evaluation framework that supports local adaptation in low-coverage settings—an essential tool for closing ZD gaps globally. The SOAMEVA evaluation was deliberately designed to assess not only what works, but also for whom and under what conditions. By combining disaggregated quantitative data with community insights and real-time feedback mechanisms, this approach offers a blueprint for more equitable monitoring and adaptive programming. Although full implementation is forthcoming, and data collection is ongoing, we believe that surfacing equity considerations from the outset enables more responsive and just immunization efforts—an essential step toward achieving IA2030 goals and Gavi’s zero-dose strategy.

## Figures and Tables

**Figure 1 vaccines-13-00834-f001:**
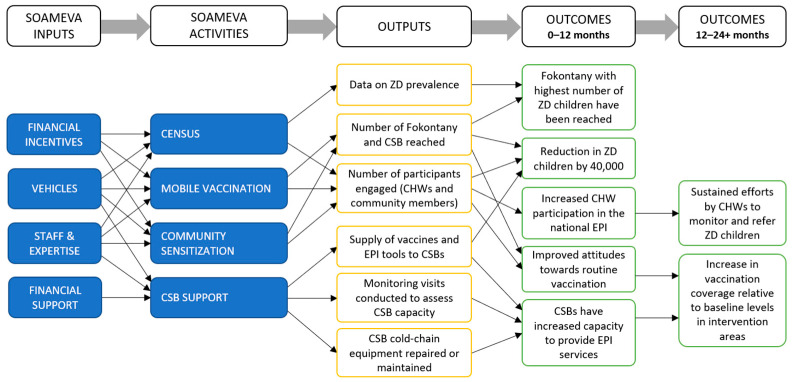
Logic model of the SOAMEVA vaccination program (2024–2026). The model shows SOAMEVA program inputs and how activities are expected to lead to improved immunization coverage and health system strengthening in Southern Madagascar. Note that long-term impacts will not be measured during this phase of the project, given its relatively short timeline.

**Figure 2 vaccines-13-00834-f002:**
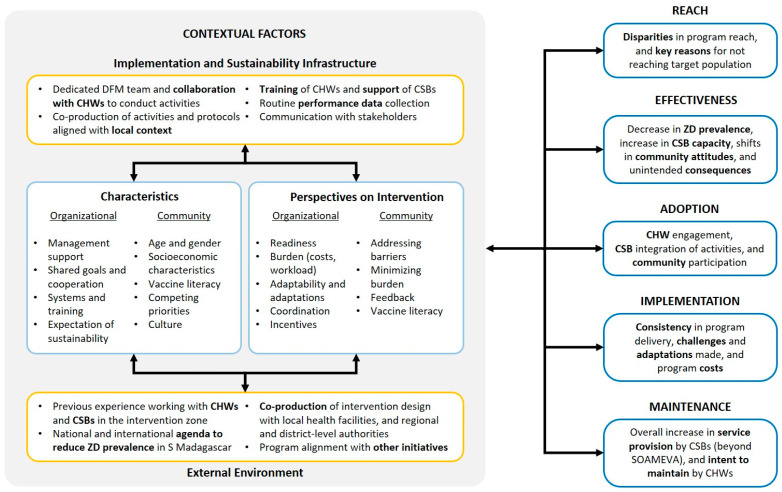
Evaluation framework of SOAMEVA, southern Madagascar, 2025, adapted from RE-AIM and PRISM [16]. The PRISM analytical lens explicitly considers contextual factors, including the implementation and sustainability infrastructure (e.g., alignment of program activities with local context), characteristics of the different actors (e.g., competing priorities), perspectives on the intervention (e.g., perceived barriers), and the external environment (e.g., the national immunization agenda). Contextual factors measured via PRISM can help understand disparities in the different RE-AIM dimensions.

## Data Availability

Not applicable.

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
