# Peer review of "Designing for Equity: An Evaluation Framework to Assess Zero-Dose Reduction Efforts in Southern Madagascar"

_vaccines, 2025, doi:10.3390/vaccines13080834_

Round 1

Reviewer 1 Report

Comments and Suggestions for Authors

This manuscript “Designing for Equity: An Evaluation Framework to Assess Zero-Dose Reduction Efforts in Southern Madagascar” presents a perspective on the design and rationale behind the equity-sensitive evaluation framework developed for the SOAMEVA program, a targeted intervention in Southern Madagascar aiming to reduce the prevalence of zero-dose (ZD) children. The authors use RE-AIM and PRISM frameworks and emphasize a mixed-methods approach that integrates disaggregated data, community feedback, and adaptive program implementation to inform policy and practice. Addressing ZD children aligns with Gavi’s Zero-Dose Strategy and the IA2030 goals. The focus on Madagascar, a country with high multidimensional poverty and low vaccine coverage, makes this highly relevant for global health equity discourse. The integration of RE-AIM and PRISM provides a well-structured lens for evaluating program performance across multiple dimensions including equity, reach, and sustainability. Embedding equity from the outset rather than treating it as an outcome is commendable and reflects modern thinking in implementation science. The use of fokontany-level data ensures fine-scale understanding of disparities, while the involvement of CHWs and caregivers supports participatory evaluation. The structured incorporation of feedback into program delivery demonstrates strong commitment to implementation responsiveness and continuous learning. The manuscript is well-written, logically organized, and effectively blends conceptual framework with practical application.

However, these aspects should be improved:

1. While this is a “Perspective” article, it would be helpful to include preliminary results or baseline indicators, if available, to enhance the practical value of the evaluation design. Please add a summary table of baseline ZD prevalence across the 16 districts or key CSB readiness indicators.

2. The term “penta-ZD” is used but could benefit from clearer, earlier definition to aid comprehension for broader audiences. Please define it explicitly in the abstract or first mention in the main text.

3. The manuscript discusses interviews and focus groups but lacks detail on sampling methods or mitigation of social desirability bias. Please briefly outline how CHW/community data were collected and analyzed to ensure objectivity.

4. While the program is context-specific, some readers may be interested in its applicability in other LMIC settings. Please add a paragraph in the discussion/conclusion on how this model could be adapted in other fragile or humanitarian contexts.

5. Figures 1 and 2 are referenced but not fully described or interpreted in the text. Please include brief figure legends within the main body or a summary sentence highlighting key elements depicted.

6. Although IRB approval is marked “Not applicable,” a brief discussion on ethical safeguards in working with vulnerable communities would be beneficial. Please add 1-2 sentences on informed verbal consent or community consent process in the data collection section.

Comments on the Quality of English Language

Minor Editing Suggestions

Line 37: “have not receive” → “have not received”

Line 90: consider simplifying the penta-ZD definition or including a citation for clarity.

Line 220 (section heading): “Reflections from the Design Stage: Evaluating for Equity in Fragile Settings” -might consider shortening for flow.

Author Response

Comment 1: While this is a “Perspective” article, it would be helpful to include preliminary results or baseline indicators, if available, to enhance the practical value of the evaluation design. Please add a summary table of baseline ZD prevalence across the 16 districts or key CSB readiness indicators.

Response 1: Thank you for this suggestion. The SOAMEVA program has now commenced, and baseline data collection is complete. However, the analysis of the baseline data is still ongoing and not yet ready for public dissemination. As this article is intended to present the evaluation design and equity-sensitive framework, rather than report results, we have opted to focus on the rationale, methodology, and anticipated analytical approach. We agree that the baseline findings will be of value, and we are preparing a follow-up manuscript focused on results and insights from the baseline analysis, which will include a detailed overview of zero-dose prevalence and CSB readiness across the 16 districts. We clarified this scope in the introduction and conclusion of the manuscript to avoid confusion about the article’s intent.

L73-74: “We focus here on the rationale, methodology, and planned analytical approach for the evaluation of the SOAMEVA program.”

L299-302: “Although full implementation is forthcoming, and data collection is ongoing, we believe that surfacing equity considerations from the outset enables more responsive and just immunization efforts – an essential step toward achieving IA2030 goals and Gavi’s Zero-Dose Strategy.”

Comment 2: The term “penta-ZD” is used but could benefit from clearer, earlier definition to aid comprehension for broader audiences. Please define it explicitly in the abstract or first mention in the main text.

Response 2: Thank you for your suggestion. We have now provided a clearer definition of penta-ZD on L37-39:

“Among these are penta-zero dose (ZD) children – zero dose being defined as “non-receipt of the first dose of the DTP-containing pentavalent vaccine (i.e. DTP1)”, which protects against diphtheria, tetanus, pertussis, hepatitis B, and Haemophilus influenzae type B [3].”

Comment 3: The manuscript discusses interviews and focus groups but lacks detail on sampling methods or mitigation of social desirability bias. Please briefly outline how CHW/community data were collected and analyzed to ensure objectivity.

Response 3: Thank you. In response, we have added the following lines L204-209 of the manuscript to clarify our qualitative sample strategy and approach to bias mitigation:

“Participants for qualitative data collection, including CHWs, caregivers, and commu-nity members, will be purposively selected to reflect diverse geographic and pro-grammatic contexts, with data collection continuing until thematic saturation is reached. To mitigate social desirability bias, data collectors will be external to both the SOAMEVA program and Madagascar health system, and will trained in neutral inter-viewing techniques. Interviews will be conducted in private settings.”  

Comment 4: While the program is context-specific, some readers may be interested in its applicability in other LMIC settings. Please add a paragraph in the discussion/conclusion on how this model could be adapted in other fragile or humanitarian contexts

Response 4: Thank you. We agree that it would be important for readers to better understand the extent to which the proposed evaluation design is applicable and relevant to other LMIC settings. As such, we added a paragraph L274-284:

“In low-to-middle income countries, under-vaccination is commonly driven by structural barriers such as geographic isolation and weak health systems [24]. Other studies investigating fragile contexts, such as Ethiopia and the Democratic Republic of Congo, have found key determinants of ZD prevalence similar to those previously identified in the Madagascar context, and have therefore emphasized the need for more equitable access to immunization services [25-27]. In addition, there is a growing body of evidence demonstrating the importance of CHW-led interventions to improve access to health services for marginalized groups in low-to-middle income countries [28-29]. While context-specific, the design of the SOAMEVA evaluation, with its core emphasis on the role of CHWs, has the potential to be adapted to other fragile settings to enhance equitable access to immunization efforts.”

Comment 5: Figures 1 and 2 are referenced but not fully described or interpreted in the text. Please include brief figure legends within the main body or a summary sentence highlighting key elements depicted.

Response 5: Thank you. We highlighted key elements depicted in Figure 1 and 2, respectively.

For Figure 1, we added the following L109-116:

“The SOAMEVA inputs include financial incentives for CHWs to conduct the community census and mobile outreach activities, the provision of vehicles for all program operations, staff to conduct the activities (e.g. training of CHWs), and financial support to strengthen the capacity of CSBs. Overall, the SOAMEVA program is expected to reach fokontany with the highest number of ZD children and reduce their prevalence in the intervention zone, both directly via mobile vaccination clinics, and indirectly by addressing community hesitancy and strengthening the capacity of CSBs to deliver immunization services, i.e. the national EPI.”

For Figure 2, we added to the following description to the Figure label:

“The PRISM analytical lens explicitly considers contextual factors, including the implementation and sustainability infrastructure (e.g. alignment of program activities with local context), characteristics of the different actors (e.g. competing priorities), perspectives on the intervention (e.g. perceived barriers), and the external environment (e.g. the national immunization agenda). Contextual factors measured via PRISM can help understand disparities in the different RE-AIM dimensions.”

Comment 6: Although IRB approval is marked “Not applicable,” a brief discussion on ethical safeguards in working with vulnerable communities would be beneficial. Please add 1-2 sentences on informed verbal consent or community consent process in the data collection section.

Response 6: While the present article does not present results from the study, as it focuses on the evaluation design, we agree these are important aspects to describe nonetheless. We included the following in the IRB and informed consent sections.

Institutional Review Board Statement: Ethics approval has been obtained from Ethics Commission of Charité - Universitätsmedizin Berlin (EA2/054/25), and the ethics application submitted to the Madagascar Ethics Evaluation Committee for Biomedical Research (CEERB) is in the final review stage and awaiting formal approval.

Informed Consent Statement: Data collection (key informant interviews and focus group discussions) is scheduled to start in September 2025. All participants will be required to be least 18 years old, and will be given detailed information (both orally and in written form) about the study. In order to take part in the study, participants will be required to sign a consent form, which states that they have received sufficient information concerning the study, and provides details on data protection measures.”

Comment 7: Line 37: “have not receive” → “have not received”

Response 7: Thank you for pointing out this typo. We have made the correction.

Comment 8: Line 90: consider simplifying the penta-ZD definition or including a citation for clarity.

Response 8: We cited the definition more explicitly for clarification.

Comment 9: Line 220 (section heading): “Reflections from the Design Stage: Evaluating for Equity in Fragile Settings” -might consider shortening for flow.

Response 9: We agree with this comment, and therefore shortened the section title to “Evaluating for Equity in Fragile Settings”.

Reviewer 2 Report

Comments and Suggestions for Authors

The paper discusses zero dose vaccination (ZD) in locations lacking in effective and effective healthcare.  Certainly not being vaccinated is a risk.  What is not mentioned is effectiveness of a single dose.  In this discussion, the importance of a single does for any vaccine can be critical.  For example, varicella single does is about 87% effective although two does is 97% (MMR for one does is 93% and 97% for a second vaccine).  The difference between a single and full vaccination schedule should be mentioned.   Importance of fully vaccinated and partially vaccination can be of great importance in preventing disease transmission. I would provide some discussion of this as part of ZD reduction. 

Author Response

Comment 1: The paper discusses zero dose vaccination (ZD) in locations lacking in effective and effective healthcare.  Certainly not being vaccinated is a risk.  What is not mentioned is effectiveness of a single dose.  In this discussion, the importance of a single does for any vaccine can be critical.  For example, varicella single does is about 87% effective although two does is 97% (MMR for one does is 93% and 97% for a second vaccine).  The difference between a single and full vaccination schedule should be mentioned.   Importance of fully vaccinated and partially vaccination can be of great importance in preventing disease transmission. I would provide some discussion of this as part of ZD reduction. 

Response 1: Thank you for pointing this out. We agree that it is important to show the difference in terms of health outcome between receiving a single dose, as opposed to completing the full vaccination schedule. Given that completing the full vaccination schedule for ZD children would certainly go beyond the SOAMEVA timeline, and may therefore depend on the health system strengthening component of the SOAMEVA program, we discussed this aspect L127-133 as follows:

 “A first dose of DTP-containing vaccine can reduce the risk of contracting life-threatening diseases such as pertussis by ca. 50% [20]. However, long-lasting im-munity for children requires completing the full vaccination schedule (i.e. three doses), which SOAMEVA aims to achieve by visiting each target fokontany three times. Ulti-mately, however, the most substantial health outcomes may only be achieved by strengthening the underlying health system and ensuring equitable and continuous access to immunization services for the most vulnerable populations.”

Reviewer 3 Report

Comments and Suggestions for Authors

Line 58. Spell it out, SOAMEVA.

Line 74. Spell it out, DFM.

Line 78. Spell it out, CHWs.

Justify the selection of the fokontany level as the smallest administrative unit in Madagascar for evaluation, as it encompasses small-scale variation in the prevalence of ZD and the program.

Line 122. The assessment is quite interesting, yet the factor of community hesitancy should also be looked at, as part of the hypothetical zero dosage.

Author Response

Comment 1: Line 58. Spell it out, SOAMEVA.

Response 1: The acronym SOAMEVA refers to the Malagasy phrase “SOrohy ny Aretina Manaova Ezaka VAksiny”, which can be translated as “Prevent disease, Make an effort to get vaccinated.” It is now spelled out L60-61.

Comment 2: Line 74. Spell it out, DFM.

Response 2: The acronym DFM is spelled out the first time it is mentioned in the manuscript, and it is included in the list of abbreviations

Comment 3: Line 78. Spell it out, CHWs.

Response 3: The acronym CHW is spelled out the first time it is mentioned in the manuscript, and it is included in the list of abbreviations

Comment 4: Justify the selection of the fokontany level as the smallest administrative unit in Madagascar for evaluation, as it encompasses small-scale variation in the prevalence of ZD and the program.

Response 4: We agree that it would be helpful to better understand the need for evaluating the intervention at the fokontany level. As such, we added the following information L190-194 concerning disparities between fokontany, based on preliminary information obtained from DFM:

“The fokontany level was selected as the unit of analysis because it is the smallest administrative unit in Madagascar with clearly defined geographic boundaries, population data, and health service linkage. This level of granularity is essential for detecting small-scale variation in zero-dose (ZD) prevalence and program reach – variation that is often masked at higher administrative levels.”

Comment 5: Line 122. The assessment is quite interesting, yet the factor of community hesitancy should also be looked at, as part of the hypothetical zero dosage.

Response 5: Thank you for pointing this out. The factor of community hesitancy will indeed be assessed as part of the evaluation, to investigate the effects of the community sensitization campaign. We now explicitly included this aspect in L109-116 where we describe Figure 1 in more detail:

“The SOAMEVA inputs include financial incentives for CHWs to conduct the community census and community-based mobile activities, the provision of vehicles to conduct all program activities, staff to conduct the activities (e.g. training of CHWs), and financial support to strengthen the capacity of CSBs. Overall, the SOAMEVA program is expected to reach the fokontany with the highest number of ZD children, to reduce the number of ZD children in the intervention zone, both directly via mobile vaccination clinics, and indirectly by reducing community hesitancy and strengthening the capacity of CSBs to deliver immunization services, i.e. the national EPI.”